# Supply Chain Finance: Cost–Benefit Differentials under Reverse Factoring with Extended Payment Terms

Hans-Martin Beyer [1,2,*] and Bodo Herzog [1,2,3,4]

1   ESB Business School, 72762 Reutlingen, Germany; Bodo.Herzog@Reutlingen-University.de
2   Economics & Finance Department, Reutlingen University, Alteburgstr. 150, 72762 Reutlingen, Germany
3   Reutlingen Research Institute (RRI), 72762 Reutlingen, Germany
4   Institute of Finance and Economics (IFE), 72762 Reutlingen, Germany
*   Correspondence: Hans-Martin.Beyer@Reutlingen-University.de

**Abstract:** This article studies the effects of reverse factoring in a supply chain when the buyer company facilitates its lower short-term borrowing rates to the supplier corporation in return for extended payment terms. We explore the role of interest rate changes, rating changes, and the business cycle position on the cost and benefit trade-off from a supplier perspective. We utilize a combined empirical approach consisting of an event study in Step 1 and a simulation model in Step 2. The event study identifies the quantitative magnitude of central bank decisions and rating changes on the interest rate differential. The simulation computes with a rolling-window methodology the daily cost and benefits of reverse factoring from 2010 to 2018 under the assumption of the efficient market hypothesis. Our major finding is that changes of crucial financial variables such as interest rates, ratings, or news alerts will turn former win–win into win–lose situations for the supplier contingent to the business cycle. Overall, our results exhibit sophisticated trade-offs under reverse factoring and consequently require a careful evaluation in managerial decisions.

**Keywords:** supply chain finance; reverse factoring; payment term extension; simulation; event study; interest rates; central bank; rating update; company news; business cycle

**JEL Classification:** G24; M21; G32; G17



## 1. Introduction

Providing financing to a business partner in a supply chain is a common phenomenon across industries. Indeed, providing trade credit downstream from the supplier to the customer is being widely used for enabling sales and a scope of other motives (Balzenko and Vandezande 2003; Seifert et al. 2013; Lam and Zhan 2021). However, financing the upstream direction, e.g., in the form of pre-payments, is less common and specific to certain industries.

In general, providing credit to a business partner implies risks and cost. The risks include the delay of due payments or in the worst case the default of the customer. Costs are implied, as the creditor has to re-finance the credit and accordingly has the cost of financing for the respective time span. The relevant capital cost depends in general on its capital structure, consisting of equity and debt. Accordingly, the costs of capital are reflected by the weighted average of these components. The larger the trade credit provided by own suppliers, the lower the capital burdens and the respective marginal interest-bearing cost. The difference between short-term assets and the trade credit from one's own suppliers is defined as net working capital (NWC). Consequently, the net working capital of a customer is reduced if the trade credit by a supplier is increased within the supply chain.

The concept of supply chain finance (SCF) implies managing financial flows in trade relationships more intelligently and at a lower cost of capital. It is well-known that supply chain finance has potential win–win situations for both parties if marginal debt financing

via a bank loan implies a higher interest burden and a lower financial flexibility for other investments (Hofmann and Belin 2011). Yet, the financial impacts of supply chain finance are dynamic and non-linear.

A scope of newer instruments enables this type of upstream financing, including Reverse Factoring (RF). Reverse factoring and supply chain finance are often used synonymously in the academic literature (Gelsomino et al. 2016). The market share of reverse factoring is approximately 3% of the entire factoring market. The global volume of factoring is about Euro 2.77 tn, according to Factors Chain International in 2018.

Reverse factoring is buyer-centric. The buyer offers a financial mechanism to suppliers, allowing a pre-financing of the suppliers' receivables at the buyer's credit conditions (Seifert et al. 2013). Consequently, to be beneficial to the supplier, such schemes require that the buyer has a better credit rating and therefore a lower interest rate than the supplier. In order to be simultaneously beneficial to the buyer, two main motivations can be identified: (i) to reduce upstream financial supply chain risks or (ii) reduce one's own costs (Klapper 2006). In practise, reducing own cost is often achieved through an extension of the buyer's payment terms with the supplier. This leads to a reduction of the buyer's and usually the supplier's NWC.

So far, the reverse factoring models are mainly examined in rather generic and stylized models in the literature. Our research focuses on reverse factoring with extended payment terms of 60–150 days on average, such as in manufacturing industries (Hofmann and Belin 2011). The extension of payment terms has the following consequences: from the buyer's perspective, the longer the trade credit terms are by the supplier, the lower one's own net working capital is. From the supplier's perspective, the lower the trade credit rates transferred from the buyer, the lower the cost of financing. Indeed, our study exhibits this pattern within a hybrid model based on empirical data and an analytical model. Among other things, we study how the role of interest rates and ratings trigger the cost–benefit allocation between the involved parties in return for extended payment terms on the buy-side (cf. Wuttke et al. 2019).

Thus, our work differs from existing literature in several dimensions: First, we simulate the cost–benefit trade-off of reverse factoring by integrating market data. We want to obtain a better understanding under which interest rate constellation reverse factoring is beneficial over the business cycle. Second, we study the impact of exogenous and endogenous factors on the cost–benefit trade-off between supply chain partners, such as central bank rates, interest rate differentials, company ratings, and company news. In order to do so, we first undertake an event study for all relevant market and company events. Third, we ultimately examine the trade-off of individual risk spreads and ratings on the cost-benefit allocation in a simulation model. Finally, we explore a break-even calculus for the supplier in order to obtain an understanding of extended payment terms on average, extending the literature by Hofmann and Belin (2011). Indeed, our research is of particular relevance in an era of ultra-low interest rates and in the aftermath of COVID-19 (Wiedmer et al. 2021; Bal and Pawlicka 2021; Gupta and Soni 2021).

We derive five major findings. First, an increasing interest rate makes reverse factoring less attractive from a supplier's perspective. Second, the difference of a single rating-category has different impacts on the cost and benefit of reverse factoring. Third, the simulation exhibits non-linearities in regard to the trade-off across rating categories. Fourth, the payment term extension from 60 to 150 days almost doubles the mean cost of reverse factoring in all constellations. Fifth, the cost of reverse factoring is lower in business cycle troughs than in booms.

Additionally, we obtain from our study two managerial conclusions: (i) The economic break-even of reverse factoring is about 80 days in comparison to credit financing over 60 days. (ii) A 100 base points increase in the interest rate or bond spread leads to a reduction of the break-even by about 5 days.

The paper is structured as follows. Section 2 provides a literature review. Section 3 explains the methodology and data. Next, in Section 4, we firstly elucidate the results of

the event study and secondly the cost–benefit dynamics over the business cycle in our simulation model. Finally, we discuss the major results including the break-even analysis in Section 5. Section 6 concludes the paper.

## 2. Literature Review

We focus the literature review on the topics related to our research, such as reverse factoring, supply chain finance, and the cost of financing.

There is a growing body of supply chain finance literature that is centered around working capital optimization. This literature examines either buyer–supplier relationships or the whole network under certain supply chain setups (Gelsomino et al. 2016; Xu et al. 2018; Wiedmer and Griffis 2021). Wetzel and Hofmann (2019) provide an overview of the working-capital literature by differentiating traditional, alternative, and progressive supply chain approaches. Seifert et al. (2013); Chakuu et al. (2019) review the trade credit literature in general. A special review on SCF and blockchain is by Liu (2021). In the following, we examine how borrowing rates and risk spreads determine reverse factoring cost and benefits.

Pfohl and Gomm (2009) characterized supply chain finance as interest rate arbitrage and considered the capital cost rate as the central starting point for optimization. They, along with Gomm (2010), highlighted the importance of payment term extensions as well as the role of rate spreads and credit ratings as relevant factors. Equally, Hofmann and Belin (2011) considered sophisticated working capital models and examined the quantitative and qualitative benefits. They found benefits including liquidity and cost savings due to lower borrowing costs, measured in the weighted average cost of capital (e.g., Lind et al. 2012; Brandenburg 2016).

Of course, there are different notions of reverse factoring in practice (Caniato et al. 2016). For instance, advanced reverse factoring is characterized by bringing several buyer's and supplier's together to increase flexibility. Wuttke et al. (2013) studied automated IT-based reverse factoring platforms. Automation integrates the data of all participants and triggers a cost-effective mechanism. This gained flexibility allows either the manual or automatic discounting of receivables with the focal buyer.

In relation to our work, Randall and Farris (2009) studied the potential of interest rate differentials in an optimal supply chain framework and found shared positive effects. Similarly, Wetzel and Hofmann (2019) analyzed in a generic interorganizational setting the win–win scenarios in supply chain financing Marchi et al. (2020). Interestingly, interest rates as such do not play an explicit role in their analysis. However, in the current low interest rate environment, it would make sense for companies with high amounts of cash to pay back earlier instead of extending payment terms with suppliers according to Wetzel and Hofmann (2019). Hence, the literature indicates a lack of studies focusing on interest rate differentials over the business cycle. Our work exactly focuses on this gap and studies the role of interest rate differentials.

Klapper (2006) examined the advantages and disadvantages of reverse factoring in detail. Indeed, she highlighted the advantages of SMEs, particularly the role of interest rate arbitrage. Similarly, Dello et al. (2015) examined the win–win situation of reverse factoring in a simulation approach and found that it depends heavily on market conditions, including interest rates. Nonetheless, they treated rates as a fixed exogenous variable. Comparably, Tanrisever et al. (2015) examined the interaction of reverse factoring on operational financial decisions. They argued that–provided the payment period remains unchanged–reverse factoring would always be preferable versus conventional debt financing from the supplier's point of view. Naturally, extended payment terms vastly influence the supplier's benefits. Related to our research question, they argued that lower risk-free rates may discourage suppliers due to higher opportunity costs in case of payment term extension. On that extent, we empirically study this notion under the present low interest rate environment.

Related to our work are studies by van der Vliet et al. (2015); Lekkakos and Serrano (2016). They explored benefits of reverse factoring, particularly from a supplier perspective. They examined payment term extensions, cost structures, information asymmetries, and other variables. Grüter and Wuttke (2017); van der Vliet et al. (2015) extended this approach to automatic and manual discounting. Manual or selective discounting leaves the supplier the choice of selling and discounting receivables in the case of cash shortages, while automatic discounting implies that all receivables are sold at the earliest opportunity. Grüter and Wuttke (2017) identified interest arbitrage as a value driver of reverse factoring. From the buyer's perspective, they examined payment term extensions and price reductions by the supplier and identified a scope of benefits to the buyer even in the case of mediocre buyer credit ratings if suppliers operate in a dynamic market. On the other hand, this implies benefits to suppliers besides the interest arbitrage via an additional liquidity option if other financing sources are constrained.[1] Consequently, strong suppliers likely reject reverse factoring because of their own attractive financing conditions (Lekkakos and Serrano 2016). New to the literature, our empirical and computational hybrid approach finds that doubling the payment terms obtains almost always direct benefits to the buyer, but less likely to the supplier.

In addition, there are studies that include financial institutions into the supply chain (Gelsomino et al. 2016; Hofmann and Zumsteg 2015). Among others, Dietrich (2012) examined the impact of a credit risk rating, the loan volume, the credit maturity, the operating cost, and the information levels on the interest rate. Economic studies focus on the role of leverage, cash, liquidity, and debt (Holmstrom and Tirole 1998; Lockhart 2014). However, several supply chain studies treat undifferentiated interest rates as a fixed exogenous variable, although credit rates depend on various factors. Overall, the literature review reinforces our research question on reverse factoring. The cost–benefit trade-offs are

- contingent on the portion of equity, debt, financial risk, and the cost of capital,
- reliant on the bank's refinancing cost, the borrower's and supplier's risk profiles, the interest rate differential, and the business cycle, and
- triggered by the difference between the size of supply chain partners.

At the same time, the literature review reveals that there is a lack of studies based on observable risk spreads, market rates, and rating spreads over the business cycle. Indeed, we exhibit the role of those variables in our combined event study and simulation approach, by focusing on marginal cost–benefit effects of interest-bearing financing.

## 3. Methodology and Data

The basic business case of a single reverse factoring transaction is driven by several variables. The time variable $t$ is determined by the credit period, i.e., the payment terms originally agreed with a bank or the buyer, including the payment term extension, in a reverse factoring scenario. The financing volume $V$ represents the receivable amounts.

Of course, the cost of capital rate is the central point of supply chain optimization. Here, the key driver is the interest rate differential between the supplier's and buyer's financial conditions. In general, it could be argued that the total interest rate differential is the only relevant variable of cost and benefit in reverse factoring. However, in a dynamic setup, we have to differentiate the components of interest rates. The major components we differentiate are

- the short-term risk-free rate $i_{RF,t}$, represented by the overnight LIBOR rate,
- the credit margins, including credit and liquidity risk $i_{BM,t}$, and
- the default risk premium, measured by credit default spreads $i_{DRP,t}^{j}$, where $j$ denotes either the buyer $B$ or supplier $S$.

Under reverse factoring, the financing volume $V$ can be pre-financed by selling the receivables to a financial institution at a discount rate that reflects the lower interest rate of the buyer $B$. Thus, the benefit of reverse factoring to the supplier is the interest rate

differential between the borrowing cost of the supplier and buyer corporation. Hence, in order to have a benchmark, we first compute the borrowing cost, *BC*, for the supplier, *S*,

$$C_{BC,t}^S = V \times \left( \frac{i_{RF,t} + i_{BM,t} + i_{DRP,t}^S}{365} \right) \times t_B, \tag{1}$$

where $t_B$ is the benchmark credit financing period of 60 days by the bank. Thus, we have to convert the annual interest rates into daily interest rates by dividing them by 365. Second, we compute the cost of a reverse factoring contract, *RF*, for the supplier, *S*, over the same period. Note that, in reverse factoring, not all steps are simultaneous, nor are they sequential. An example is the acceptance of invoice at delivery and the imminent notification of the financing institution by uploading and confirming the invoice on the platform. The delivery-to-liquidity time span usually takes a number of days, defined by $t_{Del}$. This time span may increase if the invoice is not imminently accepted or disputed by the buyer. Thus, we include a delivery-to-liquidity delay $t_{Del}$ in our model and obtain

$$C_{RF,t}^S = V \times \left[ \left( \frac{i_{RF,t} + i_{BM,t} + i_{DRP,t}^S}{365} \right) \times t_{Del} + (t_B + t_{PTE} - t_{Del}) \left( \frac{i_{RF,t} + i_{BM,t} + i_{DRP,t}^B}{365} \right) \right], \tag{2}$$

where $t_{PTE}$ denotes the payment term extension under reverse factoring. Further, we have $i_{DRP,t}^B < i_{DRP,t}^S$ and $t_B \gg t_{Del}$. The empirical range of payment term extension $t_{PTE}$ varies from industry to industry and ranges between 60 and 150 days. We simulate the cost and benefit allocation under the payment term extension of $\delta t_{PTE} = 0, 30, 60,$ and 90 days. No extension would reflect the buyer's motivation to financially stabilize the supply chain with weak suppliers.

Finally, the computation of the suppliers' cost–benefit, $CB_t^j$, is the difference of

$$CB_t^S = C_{BC,t}^S - C_{RF,t}^S, \tag{3}$$

or, in terms of percentage, we utilize $CB_t^S(\%) = (CB_t^S / C_{BC,t}^S) \times 100$. Studying the marginal impact, we neglect the fixed service provider fees of RF programs, e.g., for the onboarding participants in our model. In addition, these fees decrease with a larger number of reverse factoring schemes and transactions platforms.

In general, all these variables underlie changes over time. In particular, the risk-free rate varies due to central bank rate changes. Yet, central bank rate changes are independent from lender and creditor and therefore should be neutral to the interest rate differential as such. Nevertheless, they should not be neutral in the case of extended payment terms. The risk-related interest components are particularly dependent on credit risk related to credit ratings.

In order to provide a better understanding, we utilize our simulation model with a daily rolling window over a sample period from 2010 to 2018. Given the lack of customer-related bank and rating data and assuming the efficient market hypothesis, we focus on observable market spreads versus the risk-free rate on the one hand and the industry risk premia on the other (Table 1).

In order to explain the simulation model, we illustrate a stylized example. Suppose there is a supply chain consisting of a buyer and supplier with a financing need of 1 million EUR. For illustration purposes, we assume the risk-free rate (LIBOR) is 1% and that the bank margin is 60 bp for both costumers (Table 1); the spread for the buyer with a BBB rating is 200 bp and that for the supplier with a B+ rating is 500 bp. If the supplier takes a bank credit over 60 days, under the above assumptions, the overall cost is 9863 EUR. However, if the supplier does reverse factoring with the buyer contingent to a payment term extension to 120 days, the cost for the supplier is 10,109 EUR. Hence, reverse factoring with a 60 days payment term extension is 246 EUR, 2.5% more expensive than bank borrowing under the suppliers' credit conditions.

**Table 1.** Parameter overview of the simulation model. Source: Authors.

| Model Variables | Supplier | Buyer | Spread |
|---|---|---|---|
| **Credit Volume, $V$** | 1,000,000 EUR | - | - |
| **Cost of Financing** | | | |
| Risk-free rate, $i_{RF,t}$ | 1% | 1% | 0 |
| Bank margin, $i_{BM,t}$ | 60 bp | - | - |
| Spread Supplier-Buyer, $\delta i^j_{DRP,t}$ | 500 bp | 200 bp | $-300$ bp |
| Service provider fee, $F$ | 0 EUR | 0 EUR | 0 |
| **Payment Terms** | | | |
| Conventional Bank Financing, $t_B$ | 60 days | - | - |
| Payment Term Extension, $\delta t_{PTE}$ | 60 days | - | - |
| Liquidity delay $t_{Del}$ | 3 days | - | - |
| **Simulation Output** | | | |
| Total Cost of Conventional Financing, $C^S_{BC,t}$ | 9863.01 EUR | - | - |
| Total Cost of Reverse Factoring, $C^S_{RF,t}$ | 10,109.59 EUR | - | - |
| Cost–Benefit Delta, $CB_s$ | 246.50 EUR | - | - |
| **Cost–Benefit in percent, $CB^S_t$** | +2.5% | - | - |

In general, we utilize this simulation model, and extend it to a dynamic setup. Thereafter, we simulate the cost–benefit trade-off each day with a rolling-window over the time period from 2010 to 2018. As inputs, we use observable market spreads in order to account for the empirical realities due to interest rate dynamics and rating changes. Those changes are derived from our event study. Hence, the final output of the simulation model is a daily time-series of potential cost–benefit effects in percentage from the supplier's perspective. A positive percentage number denotes additional costs of reverse factoring in comparison to traditional credit financing via a financial intermediary.

Our unique data consist of daily observations, comprising 70 corporate bonds of US corporations in the non-durable consumer goods sector, including some automotive related firms, over the period from 4 January 2010 to 7 March 2018. The firms are summarized in Table A1 in the appendix. Overall, the sample includes around $N = 570,000$ observations. Thus, our data extends well beyond the sample size of similar studies. To our knowledge, this is one of the first studies that simulates potential cost and benefits of reverse factoring in regard to exogenous factors over the business cycle. All data were retrieved from FactSet on 7 March 2018. Figure 1 illustrates the daily corporate bond spreads of the selective firms in regard to the mean value of each rating category in our sample.

As expected, the mean spreads of corporate bonds with an A-rating is the lowest (green curve). Noteworthy, the difference between the mean of B-ratings and BB-ratings is not conclusive over the time period. In the first years of the sample, BB-rated firms had higher spreads than B-rated assets. This observation reflects the financial turmoil in the aftermath of the global financial crisis from 2008 to 2009 (Reinhard and Rogoff 2009). Subsequently, the bond spreads continuously decline to levels around 100–250 at the end of our sample period in 2018. In that regard, our sample contains different economic environments. This is required in order to comprehensively assess the cost and benefit trade-off of reverse factoring over the business cycle.

Note that our model distinguishes from existing research by utilizing observed market spreads between suppliers and buyers. We use corporate bond spreads for two reasons: (i) the availability of data. Indeed, there is no internal rating data and commercial bank credit spread data available, except some anecdotal evidence. (ii) Measuring bond spreads is a common proxy in the literature.

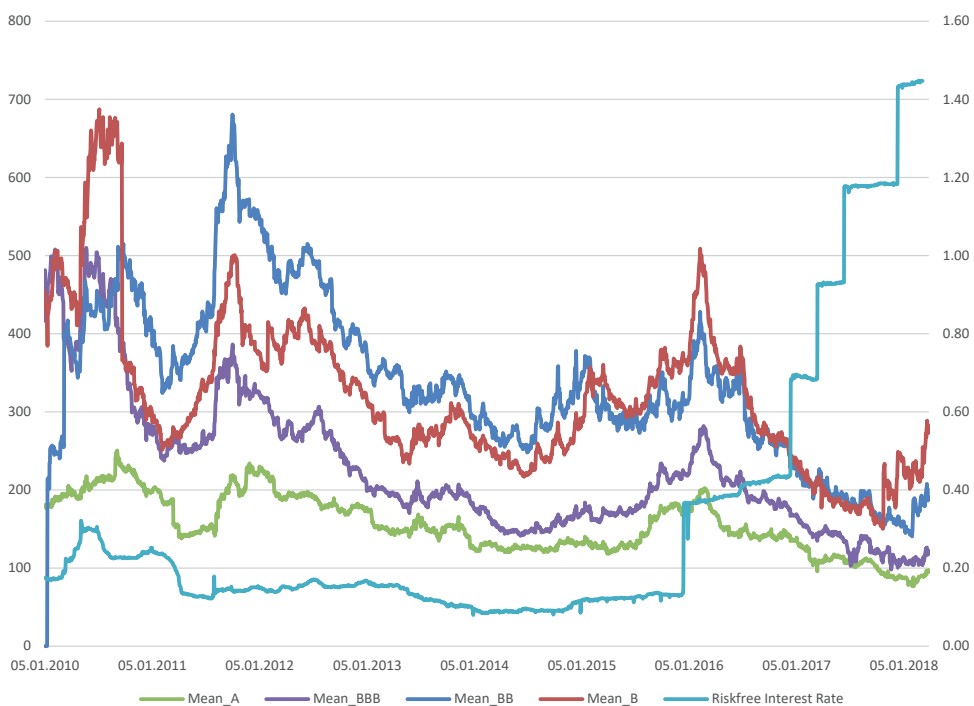

**Figure 1.** Bond spreads of credit rating categories over time (non-durables). The figure denotes the mean of A, BBB, BB, and B Spreads (category consumer non-durables; left-hand scale) and the risk-free interest rate (right-hand scale). Source: Authors, FactSet.

In sum, our analysis consists of two steps. In Step 1, we identify via an event study the magnitude of changes on the spreads by three different events: (i) central bank rate changes, (ii) rating updates, and (iii) company news alerts. In Step 2, we utilize this information in our simulation in order to compute the daily cost and benefit trade-off of reverse factoring for differently rated firms over the sample period.

## 4. Two-Stage Simulation Model

Our simulation model and our event study methodology together with the empirical data allow us to quantitatively simulate the cost and benefit trade-off of reverse factoring.

### 4.1. Event Study

At first, we exercise our event study. In order to study the role of changes in interest rates and bond spreads across differently rated firms, we identify representative benchmarks. Causal identification is not straightforward because different news or rating upgrades might have different impacts over time. Our event study distinguishes three channels. The direct interest rate changes in response to the central banks. Changes in the refinancing rates occur due to rating updates, and changes in refinancing rates occur due to company news. In order to identify the absolute magnitude of those events, we utilize ex-ante and ex-post event windows of 12 days.

In general, in the finance literature, an event study is particularly used to assess news events (Fama et al. 1969; see MacKinlay 1997 for a survey). Equally, there is research work identifying structural shocks from historical events (see, among others, Brown and Warner 1985; Romer and Romer 2004; Ramey 2011; Stock and Watson 2012; Gertler and Karadi 2015). We follow those event studies in order to identify the magnitudes on interest rate spreads of the three different event categories. Consequently, this part of the analysis generates its own unique empirical findings.

The identification of the events is critical. We selected all key historical events by central banks, rating agencies, and the 70 corporations in our sample period. First, we

identified 177 central bank events. Second, we studied 22 rating up- or downgrades. Third, we examined 1262 company news events.

Based on the event window, we computed the mean impact of the event on the corporate bonds in each of the three categories. We separated positive and negative events, e.g., by grouping events, defined by a higher or lower spread after the event date. By this grouping, we were able to compute the mean positive and negative impact on bonds of each event. Table 2 summarizes the results of the event study.

**Table 2.** Summary of the event study. Authors' computations.

| Category | Negative Events | Positive Events | All Events |
|---|---|---|---|
| Central Bank, $N = 177$ | 49.7 bp | −22.6 bp | 6.5 bp |
| Rating Updates, $N = 22$ | 17.3 bp | −17.5 bp | −3.9 bp |
| Company News, $N = 1262$ | 26.4 bp | −24.6 bp | −1.8bp |

Note: The sample period is from 2010 to 2018 and comprises 70 companies. The pre- and post-event window is 12 days. *bp* denotes basis points. The numbers denote the change in the interest or bond spread in basis points for the events. We distinguish (i) central bank events, (ii) rating events, and (iii) company news events. In each category, we distinguish positive and negative events. A positive event is defined as a cut of the interest rate, a rating upgrade, or positive company news. These events lower the spreads (negative sign). The opposite is defined as a negative event. The category 'All Events' is the average of positive and negative events. We computed the difference between corporate spreads and the LIBOR rate. The LIBOR is our proxy for the risk-free rate.

In the case of a negative event, the bond spread increases on average by 17 basis points after a rating downgrade and by 26.4 basis points after negative company news. In the case of positive events, we identify a similar or roughly symmetric magnitude. For central bank events, we obtain a rise of spreads on average by 49.7 basis points. Note that, since December 2015, the FED has started to raise rates by 25 basis points in almost every policy meeting until our sample end in 2018. On the contrary, the monetary (and fiscal) stabilization reduced bond spreads on average by 22.6 basis points. The central bank events on corporate bonds are asymmetric. The column 'All Events' is the average of all positive and negative events. Thus, the magnitude mitigates per definition.

*4.2. Simulation Model: Cost and Benefit Trade-Off*

In Step 2, we simulate the cost and benefit trade-off to a typical reverse factoring transaction. Each simulation utilizes one of the benchmark spreads obtained by the event study above. We normalize the credit amount to 1 million EUR over a period of 60 days and analyze the impact of extended payment terms. For each constellation, we compute the relative cost or benefit in percentage. Furthermore, we distinguish between central bank, rating, and company events over the business cycle. Overall, the simulation gives us, for each of our corporations, a daily time-series of the cost–benefit trade-off in percentage (cf. Figure 2).

Figure 2 represents a single graphical simulation output with the cost–benefit trade-off in percentage of a reverse factoring contract from the suppliers' perspective with payment terms of 120 days in comparison to a standard credit scenario of 60 days. The simulation incorporates, on the one hand, the daily observations of the corporate bonds and, on the other, the daily interest rate spread over the sample period from 2010 to 2018.

Under our assumptions, the interest rate differential only indicates a benefit of reverse factoring of up to 40% in the period from 2013 through 2016. In the period from 2010 to 2012 and after 2016, our analysis displays a significant disadvantage of up to 100% for the supplier (Figure 2). Naturally, there is a daily variation of costs and benefits based on the market volatility of interbanking rates and the macro-economic development. At the end of the sample period, the cost of reverse factoring is more expensive than credit financing in a range from 40 to 60%. Consequently, the daily simulation allows us to derive break-even points, at which reverse factoring is more expensive than credit financing. The next section discusses the results in detail.

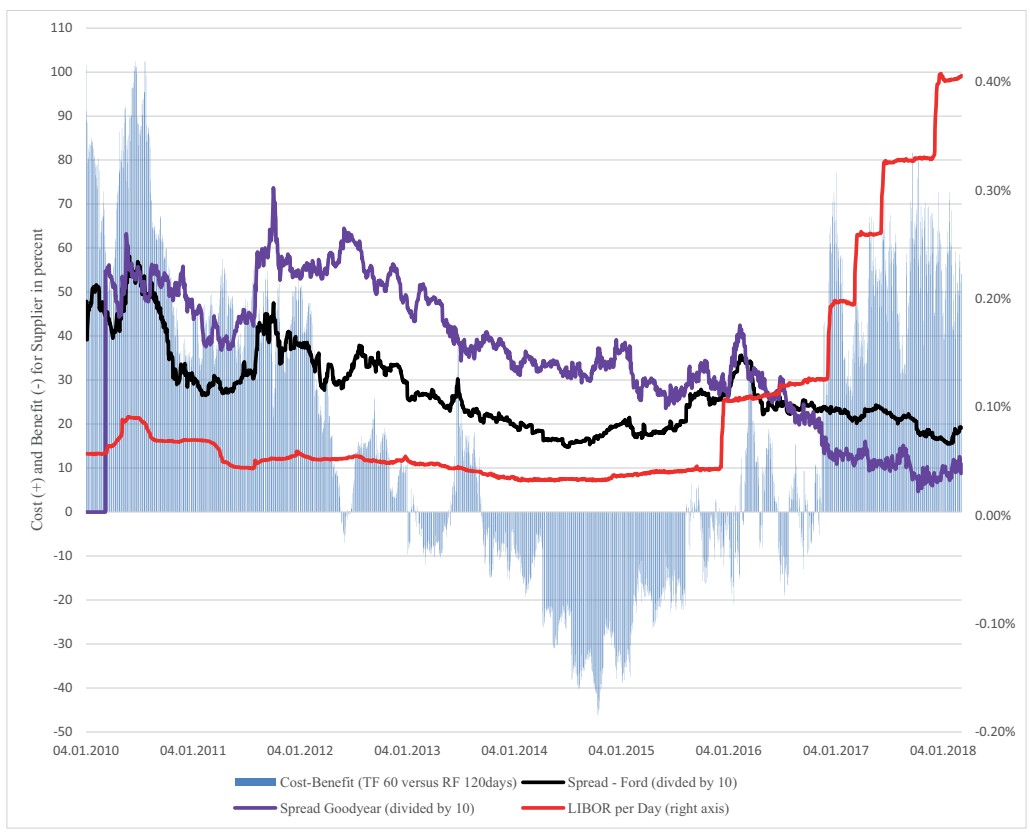

**Figure 2.** Simulation of cost and benefit trade-off in percent over the business cycle. The figure denotes the cost of reverse factoring in percent (60 vs. 120 days) in comparison to bank credit financing over 60 days (customer: Ford; supplier: Goodyear). Source: Authors' simulation.

## 5. General Results and Discussion

The main objective is to analyze the impact of interest rates, ratings, and news alerts as well as the extended payment terms on the cost–benefit allocation from the suppliers' perspective.

First, in order to explain the output, we start with a stylized simulation of raising central bank interest rates by 100 basis points. Table 3 summarizes the simulation result. Each number is separately computed for a given payment term extension and a rating constellation across buyers and suppliers. For instance, the number of −15.09% denotes that reverse factoring is 15.09% cheaper than a credit to the supplier over 60 days, where the customer has an A-rating and the supplier has a BBB-rating (Table 3). The benefit is obvious because the A-rating of the buyer facilitates the lower credit rates to the supplier over the period of 60 days. Therefore, the buyer demands a payment term extension from 60 days to 90 or 120 days in order to obtain an equal benefit by a reverse factoring contract. An extended payment term to 90 days makes reverse factoring on average 26.97% more expensive than a bank credit to the supplier. Yet, the buyer benefits now from the delayed payment. Consequently, the longer the extension of payment terms, the lower the benefits to the supplier, and the higher the benefits to the buyer. This pattern is visible over all rating constellations and rate adjustments.

**Table 3.** Stylized simulation of 100 bp higher interest rates. Source: Authors' simulation.

| Timeseries 2010 to 2018 | | Cost-Benefit of Reverse Factoring in Response to **Standardized Interest Rate** | | | | | |
|---|---|---|---|---|---|---|---|
| | | All Firms of Consumer Durables | | | | Special Cases | |
| **in percent** Cost (+) or Benefit (-) | | Rating A vs Rating BBB | Rating A vs Rating BB | Rating BBB vs Rating BB | Rating BBB vs Rating B | Automotive | Example |
| Benchmark | RF 60 vs 60 | -15.09 | -32.13 | -19.46 | -17.58 | -6.21 | -30.32 |
| | RF 60 vs 90 | 26.97 | 0.96 | 20.30 | 23.17 | 40.52 | 3.72 |
| | RF 60 vs 120 | 69.03 | 34.05 | 60.06 | 63.92 | 87.25 | 37.76 |
| | RF 60 vs 150 | 111.09 | 67.13 | 99.82 | 104.67 | 133.99 | 71.80 |
| Interest rate (credit spread) **increases** by 100 basis points | RF 60 vs 60 | -12.29 | -27.03 | -16.37 | -14.64 | -5.12 | -25.48 |
| | RF 60 vs 90 | 31.24 | 8.74 | 25.02 | 27.65 | 42.18 | 11.10 |
| | RF 60 vs 120 | 74.78 | 44.52 | 66.40 | 69.95 | 89.49 | 47.69 |
| | RF 60 vs 150 | 118.31 | 80.29 | 107.79 | 112.24 | 136.79 | 84.28 |
| Interest rate (credit spread) **decreases** by 100 basis points | RF 60 vs 60 | -19.66 | -39.70 | -24.03 | -22.05 | -7.91 | -37.60 |
| | RF 60 vs 90 | 19.99 | -10.59 | 13.32 | 16.35 | 37.92 | -7.39 |
| | RF 60 vs 120 | 59.65 | 18.51 | 50.68 | 54.75 | 83.75 | 22.83 |
| | RF 60 vs 150 | 99.30 | 47.62 | 88.03 | 93.14 | 129.59 | 53.04 |
| No. | | 6 (A) 59 (BBB) | 6 (A) 21 (BB) | 59 (BBB) 21 (BB) | 59 (BBB) 13 (B) | 4 (Producer) 16 (Supplier) | 1 (Ford, BBB) 1 (Goodyear, BB) |

Notes: This Table reports the cost (positive sign) and benefits (negative sign) of reverse factoring from the point of view of a supplier corporation. The computation assumes a benchmark loan volume of 1.000.000 Euro. The profit margin of banks is assumed to be 50 basis points. The central bank interest rate is assumed at 1 percent. The computation is based on unique daily timeseries from 2010 to 2018 for each of 70 US-corporations in the automotive sector (category consumer non-durables; FactSet). The last row represents the number of corporations in the different rating categories in our data set. 'RF' stands for Reverse Factoring. The reading of the Table is as follows: The row 'Benchmark' characterizes a benchmark constellation with longer maturities of reverse factoring contracts. The supplier needs a loan of 60 days but the originator (producer) offers a reverse factoring contract with longer maturities. The RF payment terms are ranging from 60 days (RF 60 vs 60) up to a maturity extension of 150 days (RF 60 vs 150). The other rows reflect the cost-benefit trade-off for the average, positive, negative events of an interest change by the central bank. The columns represent the rating constellation of both corporations making the RF contract. The column 'Rating A vs Rating BBB' denotes a situation where we have 59 suppliers with a BBB-rating and 6 originators with an A-rating. The number of -19.66 represents a benefit of -19.66 percent or, in other words, the cost saving of reverse factoring in comparison to a direct credit financing to the supplier corporation. The cost saving is due to the offer of a lower loan rate by the customer. The benefits turn into costs, if the customer (the contractor) requires a maturity extension of the loan payment. For instance, in case of RF 60 vs 150, the number is of 99.30. It denotes that the reverse factoring contract is of 99.30 percent more expensive (due to maturity extension) than a direct credit financing over just 60 days to the supplier corporation by the supplier borrowing rate. Source: Authors computations.

We examine five findings in general. First, an increasing interest rate makes reverse factoring less attractive or, in other words, more expensive from the suppliers' perspective. Unsurprisingly, this pattern is robust in all situations. Second, the difference of a single rating category, such as an *A* versus *BBB* rating, has different impacts on the cost and benefits of reverse factoring. While a payment term extension from 60 to 90 days is 26.97% more expensive for a *A*- versus *BBB*-rated firm, it is only 20.30% for *BBB*- versus *BB*-rated firms. This difference is sizeable. Third, the simulation examines non-linearities in regard to the cost and benefits across rating categories due to the observed non-linear patterns in corporate bond dynamics. Fourth, a payment term extension from 60 to 150 days increases the mean cost of reverse factoring by approximately 100% in almost all constellations. Fifth, the cost of reverse factoring is slightly lower in business cycle troughs than booms (Appendix D—Figure A2). Indeed, under certain conditions, reverse factoring is even cheaper for a payment term extension to 90 days. For instance, if the buyer corporation has a *A* rating, then, on average, reverse factoring is −10.59% cheaper than credit financing.

Next, we compute the impact of reverse factoring based on our event study benchmarks. For the central bank case, the result is in Table 4. The output of the other simulation exercises are reported in Appendix B (Tables A2 and A3). Two findings stand out: First, the magnitude on cost and benefit is lower under the observed market conditions. Second, the direction and pattern of cost and benefit are comparable to the stylized case above. Surprisingly, we find a kind of symmetry in ratings and news events in our sample.

**Table 4.** Simulation event study—adaption to central banks' interest rates. Source: Authors' simulation.

| Timeseries 2010 to 2018 | | Cost-Benefit of Reverse Factoring in Response to real **Central Bank Decisions** | | | | | |
|---|---|---|---|---|---|---|---|
| | | All Firms of Consumer Durables | | | | Special Cases | |
| **in percent** Cost (+) or Benefit (-) | | Rating A vs Rating BBB | Rating A vs Rating BB | Rating BBB vs Rating BB | Rating BBB vs Rating B | Automotive | Example |
| Benchmark | RF 60 vs 60 | -19.66 | -39.70 | -24.03 | -22.05 | -7.91 | -37.60 |
| | RF 60 vs 90 | 19.99 | -10.59 | 13.32 | 16.35 | 37.92 | -7.39 |
| | RF 60 vs 120 | 59.65 | 18.51 | 50.68 | 54.75 | 83.75 | 22.83 |
| | RF 60 vs 150 | 99.30 | 47.62 | 88.03 | 93.14 | 129.59 | 53.04 |
| Interest rate (credit spread) **increases** by 6.5 basis points (mean event between 2010-2018) | RF 60 vs 60 | -19.27 | -39.10 | -23.67 | -21.69 | -7.77 | -37.01 |
| | RF 60 vs 90 | 20.58 | -9.67 | 13.88 | 16.90 | 38.13 | -6.49 |
| | RF 60 vs 120 | 60.44 | 19.75 | 51.42 | 55.49 | 84.04 | 24.03 |
| | RF 60 vs 150 | 100.29 | 49.17 | 88.97 | 94.07 | 129.95 | 54.54 |
| Interest rate (credit spread) **decreases** by 22.6 basis points (all negative events) | RF 60 vs 60 | -21.14 | -41.95 | -25.39 | -23.40 | -8.45 | -39.79 |
| | RF 60 vs 90 | 17.73 | -14.03 | 11.25 | 14.28 | 37.11 | -10.74 |
| | RF 60 vs 120 | 56.61 | 13.89 | 47.89 | 51.96 | 82.67 | 18.32 |
| | RF 60 vs 150 | 95.48 | 41.80 | 84.52 | 89.64 | 128.22 | 47.37 |
| Interest rate (credit spread) **increases** by 49.7 basis points (all positive events) | RF 60 vs 60 | -17.07 | -35.53 | -21.51 | -19.57 | -6.96 | -33.57 |
| | RF 60 vs 90 | 23.95 | -4.23 | 17.17 | 20.14 | 39.38 | -1.24 |
| | RF 60 vs 120 | 64.96 | 27.08 | 55.85 | 59.84 | 85.71 | 31.09 |
| | RF 60 vs 150 | 105.98 | 58.38 | 94.53 | 99.54 | 132.05 | 63.42 |
| No. | | 6 (A) 59 (BBB) | 6 (A) 21 (BB) | 59 (BBB) 21(BB) | 59 (BBB) 13 (B) | 4 (Producer) 16 (Supplier) | 1 (Ford, BBB) 1 (Goodyear, BB) |

Notes: This Table reports the cost (positive sign) and benefits (negative sign) of reverse factoring from the point of view of a supplier corporation. The computation assumes a benchmark loan volume of 1.000.000 Euro. The profit margin of banks is of 50 basis points.The central bank interest rate is assumed to be zero. The computation is based on daily timeseries from 2010 to 2018 for 70 US-corporations in the automotive sector. The last row represents the number of corporations in the rating categories. The reading of the Table is as follows: The row 'Benchmark' characterizes a benchmark constellation with different maturities of reverse factoring contracts. The supplier needs a loan of 60 days but the customer (producer) offers a reverse factoring contract with different maturities. The maturity is ranging from 60 days (RF 60 vs 60) up to 150 days (RF 60 vs 150). The other rows reflect the cost-benefit trade-off after positive or negative interest rate changes by the central bank. All events (interest rate changes) are based on the event study methodology. The column 'Rating A vs Rating BBB' denotes a situation where we have 59 suppliers with a BBB-rating and 6 originators with an A-rating. The number of -19.66 represents a benefit of -19.66 percent or, in other words, the cost saving of reverse factoring in comparison to a bank financing by a supplier corporation. For instance, in case of RF 60 vs 150, the number is of 99.30. It denotes that the reverse factoring contract is of 99.30 percent more expensive due to payment term extension. Source: Authors computations.

Finally, in order to derive some managerial policy conclusions, we compute a break-even day at which the benefits of reverse factoring turn into costs on average. In our benchmark scenario, the break-even is approximately 80 days (Appendix C—Figure A1). Table 5 summarizes the break-even analysis. We find that the break-even day varies with the rating constellation. Indeed, we observe, for certain constellations, a declining break-even by approximately five days. Moreover, the slope of the break-even line is positive; however, the difference across rating categories is insignificant (Appendix C—Figure A1). Furthermore, we observe a level effect, particularly for the constellation of firms with *A* versus *BB* ratings. This corroborates the multifaceted determinants and non-linearities of reverse factoring trade-offs.

Based on the break-even analysis, we derive two managerial conclusions: (i) The economic break-even of reverse factoring is about 80 days contingent to credit financing over 60 days. (ii) Increases in interest rates or bond spreads by 100 basis points lead to a reduction in the break-even by about 5 days. Indeed, ceteris paribus, the result exhibits that a downgrading of the buyer negatively correlates to the suppliers cost of reverse factoring. Both rules of thumb are practical guidelines to assess the cost and benefit trade-off on reverse factoring.

In that context, our hybrid model obtains several interesting results for discussion. The payment term extensions are to the disadvantage of suppliers depending on the pass-through of interest rate changes by the financial intermediary. Not surprisingly, the larger the rating or interest rate differential, the smaller the disadvantage. Furthermore, in the aftermath of the global financial crisis, we find that the lower central bank interest rates did not have an overall positive impact to the supplier company. Yet, the low rates have

cost implications on suppliers via the rating downgrades of financially weak supply chain partners. Indeed, financially weaker companies are strongly impacted in their ability to obtain credit from intermediaries or capital markets. On the one hand, the general business proposition of accepting an extension of payment terms is a risky proposition for suppliers. On the other hand, in times of economic and financial turmoil, reverse factoring serves as a facilitator and stabilizer in the supply chain.

**Table 5.** Summary of break-even analysis.

| Break-Even Points Rating Constellation | Scenario I in Days | Scenario I + 100 bp in Days | Difference in Days |
|---|---|---|---|
| A vs. BBB | 75 | 71 | 4 |
| A vs. BB | 101 | 90 | 11 |
| BBB vs. BB | 80 | 75 | 5 |
| BBB vs. B | 78 | 73 | 5 |

Note: Scenario I assumes a loan volume of 1,000,000 EUR, a zero central bank rate, and a bank margin of 50 bp. Source: Authors' computations.

We are aware of the limitations despite our substantial data. One restriction is the data period and unequal bond maturities. The special period after the great recession of 2008 probably contains anomalies. A major issue is the highly accommodative and unconventional monetary policy during that period. Moreover, we focused mainly on corporations in the non-durable consumer sector, particularly the automotive sector, because this sector commonly utilizes reverse factoring (Lampón et al. 2021). Hence, we cannot generalize the findings to the overall economy.

Our study neglects the indirect effects on the overall average cost of capital. However, our rating data might reflect capital structure effects, and any change in the overall risk-free interest rate level does not affect the interest rate differential, according to the theory of the capital asset pricing model (CAPM), despite the implications on the absolute level of capital cost.

The methodological challenge is twofold. On the one hand, the estimation of short-term bank credit spreads from longer-term bond spreads. On the other hand, the identification of the interest rate pass-through affecting commercial customers. We are likely underestimating the real effects due to lower risk premia in shorter-term rates according to a normal yield curve. Yet, Dietrich (2012) as well as anecdotal evidence support our proxies. Nonetheless, extending our analysis in light of the limitations is a task for future research.

## 6. Conclusions

The paper provides a novel vantage point to the supply chain finance literature in a world of low interest rates. We developed a combined event study and a simulation model that exhibit new insights on the fundamental question of the cost–benefit trade-off of reverse factoring in various economic environments. Indeed, little is known about the cost–benefit trade-off based on the present research in regard to changes in central bank rates, changes in ratings, news alters and how the business cycle is affecting reverse factoring.

We demonstrate that changes in financial variables, such as a rising interest rate or rating, will turn a former win–win situation into a win–lose situation for suppliers, despite lower financing rates under reverse factoring. The same might apply in times of an economic downturn if the credit rating of the originator, i.e., the buyer, deteriorates. In addition, the pass-through of lower policy rates takes time, while the sudden adjustment of market spreads leads, in the worst case, to an abrupt stop of credit line prolongation to the supplier's liquidity. Consequently, accepting or rejecting a payment term extension is a sophisticated managerial task for independently acting suppliers.

**Author Contributions:** Both authors equally contributed to this research. All authors have read and agreed to the published version of the manuscript.

**Funding:** We are grateful that the publication fee was financed by Albert-Ludwigs-Universität Freiburg, Universitätsbibliothek, Geschäftsstelle des Konsortiums Baden-Württemberg. We have not received further external funding for the research project.

**Institutional Review Board Statement:** Not applicable.

**Informed Consent Statement:** Not applicable.

**Data Availability Statement:** All data is available.

**Acknowledgments:** We thank conference participants for feedback and comments on a preliminary version and the anonymous referees for excellent feedback and comments on the paper. All remaining errors are our responsibility.

**Conflicts of Interest:** The authors declare no conflict of interest.

## Abbreviations

The following abbreviations are used in this manuscript:

| | |
|---|---|
| SCF | Supply Chain Finance |
| RF | Reverse Factoring |
| NWC | Net Working Capital |
| $B$ | Buyer Coporation |
| $S$ | Supplier Coporation |
| $V$ | Financing Volume of 1 Mio. |
| $CB$ | Cost–benefit |
| $BC$ | Borrowing cost |
| $BM$ | Bank margin |
| $bp$ | Basis points |
| $DRP$ | Default risk premium |
| $Del$ | Delivery-to-liquidity delay |
| $PTE$ | Payment terms extension |
| $LIBOR$ | London Interbank Offered Rate |
| $t_{BM}$ | Benchmark borrowing period of 60 days |
| $t_{Del}$ | Pre-financing delay of 3 days |
| $t_{PTE}$ | Payment Terms Extension of $0, 30, 60$, and 90 days |
| $i_{RF,t}$ | Risk-free interest rate at time $t$ |
| $i_{BM,t}$ | Bank Margin Rate at time $t$ |
| $i_{DRP,t}$ | Default Risk Premium at time $t$ |

## Appendix A. List of Companies

**Table A1.** List of companies. Source: Authors; FactSet data source.

| No. | Company Names |
|---|---|
| 1 | Stanley Black & Decker, Inc. |
| 2 | Volkswagen Group of America Finance LLC |
| 3 | Ford Motor Company |
| 4 | General Motors Company |
| 5 | D.R. Horton, Inc. |
| 6 | Whirlpool Corporation |
| 7 | Electronic Arts Inc. |
| 8 | Hasbro, Inc. |
| 9 | Activision Blizzard, Inc. |
| 10 | Leggett & Platt, Incorporated |
| 11 | NVR, Inc. |
| 12 | Post Apartment Homes LP |
| 13 | Mohawk Industries, Inc. |
| 14 | Harman International Industries, Incorporated |
| 15 | Snap-on Incorporated |

**Table A1.** *Cont.*

| No. | Company Names |
|---|---|
| 16 | Harley-Davidson, Inc. |
| 17 | Daimler North America Corp. |
| 18 | M.D.C. Holdings, Inc. |
| 19 | Lennar Corporation |
| 20 | Goodyear Tire & Rubber Company |
| 21 | TRI Pointe Group, Inc. |
| 22 | Meritage Homes Corporation |
| 23 | PulteGroup, Inc. |
| 24 | Meritage Homes Corporation |
| 25 | Mattel, Inc. |
| 26 | KB Home |
| 27 | Monarch Communities, Inc. |
| 28 | BorgWarner Inc. |
| 29 | Lear Corporation |
| 30 | Dana Incorporated |

## Appendix B. Simulation Results

**Table A2.** Simulation results with rating events. Source: Authors, FactSet data source.

| Timeseries 2010 to 2018 | | Cost-Benefit of Reverse Factoring in Response to **Rating News Decisions** | | | | | |
|---|---|---|---|---|---|---|---|
| | | All Firms of Consumer Durables | | | | Special Cases | |
| **in percent**<br>Cost (+) or Benefit (-) | | Rating A vs<br>Rating BBB | Rating A vs<br>Rating BB | Rating BBB<br>vs Rating BB | Rating BBB<br>vs Rating B | Automotive<br>Sector | Example |
| Benchmark | RF 60 vs 60 | -19.66 | -39.70 | -24.03 | -22.05 | -7.91 | -37.60 |
| | RF 60 vs 90 | 19.99 | -10.59 | 13.32 | 16.35 | 37.92 | -7.39 |
| | RF 60 vs 120 | 59.65 | 18.51 | 50.68 | 54.75 | 83.75 | 22.83 |
| | RF 60 vs 150 | 99.30 | 47.62 | 88.03 | 93.14 | 129.59 | 53.04 |
| Interest rate (credit spread) **decreases** by 3.9 basis points (mean event between 2010-2018) | RF 60 vs 60 | -19.90 | -40.07 | -24.25 | -22.27 | -8.02 | -38.03 |
| | RF 60 vs 90 | 19.63 | -11.16 | 12.98 | 16.01 | 37.76 | -8.05 |
| | RF 60 vs 120 | 59.15 | 17.75 | 50.22 | 54.29 | 83.54 | 21.93 |
| | RF 60 vs 150 | 98.68 | 46.66 | 87.45 | 92.57 | 129.32 | 51.92 |
| Interest rate (credit spread) **decreases** by 17.5 basis points (all negative events) | RF 60 vs 60 | -20.79 | -41.42 | -25.07 | -23.08 | -8.43 | -39.75 |
| | RF 60 vs 90 | 18.27 | -13.22 | 11.74 | 14.77 | 37.13 | -10.68 |
| | RF 60 vs 120 | 57.34 | 14.98 | 48.54 | 52.62 | 82.69 | 18.40 |
| | RF 60 vs 150 | 96.40 | 43.18 | 85.35 | 90.47 | 128.25 | 47.48 |
| Interest rate (credit spread) **increases** by 17.3 basis points (all positive events) | RF 60 vs 60 | -18.67 | -38.14 | -23.09 | -21.11 | -7.61 | -36.32 |
| | RF 60 vs 90 | 21.51 | -8.21 | 14.76 | 17.78 | 38.39 | -5.44 |
| | RF 60 vs 120 | 61.68 | 21.72 | 52.61 | 56.66 | 84.38 | 25.45 |
| | RF 60 vs 150 | 101.86 | 51.65 | 90.46 | 95.55 | 130.38 | 56.33 |
| No. | | 6 (A)<br>59 (BBB) | 6 (A)<br>21 (BB) | 59 (BBB)<br>21(BB) | 59 (BBB)<br>13 (B) | 4 (Producer)<br>16 (Supplier) | 1 (Ford, BBB)<br>1 (Goodyear, BB) |

Notes: This Table reports the cost (positive sign) and benefits (negative sign) of reverse factoring from the point of view of a supplier corporation. The computation assumes a benchmark loan volume of 1.000.000 Euro. The profit margin of banks is assumed to be 50 basis points. The central bank interest rate is assumed to be zero. The computation is based on daily timeseries from 2010 to 2018 for 70 US-corporations in the category consumer non-durables. The last row represents the number of corporations in the rating categories. The reading of the Table is as follows: The row 'Benchmark' characterizes a benchmark constellation with different maturities of reverse factoring contracts. The supplier needs a loan of 60 days but the customer (producer) offers a reverse factoring contract with longer maturities. The payment terms is ranging from 60 days (RF 60 vs 60) up to 150 days (RF 60 vs 150). The other rows reflect the cost-benefit trade-off. All events are based on the event study. The column 'Rating A vs Rating BBB' denotes a situation where we have 59 suppliers with a BBB-rating and 6 customer with an A-rating. The number of -19.66 represents a benefit of -19.66 percent or, in other words, the cost saving of reverse factoring to the supplier. Source: Authors computations.

**Table A3.** Simulation results with corporate news events. Source: Authors, FactSet data source.

| Timeseries 2010 to 2018 | | Cost-Benefit of Reverse Factoring in Response to **Company News Decisions** | | | | | |
|---|---|---|---|---|---|---|---|
| | | All Firms of Consumer Durables | | | | Special Cases | |
| **in percent**<br>Cost (+) or Benefit (-) | | Rating A vs Rating BBB | Rating A vs Rating BB | Rating BBB vs Rating BB | Rating BBB vs Rating B | Automotive Sector | Example |
| Benchmark | RF 60 vs 60 | -19.66 | -39.70 | -24.03 | -22.05 | -7.91 | -37.60 |
| | RF 60 vs 90 | 19.99 | -10.59 | 13.32 | 16.35 | 37.92 | -7.39 |
| | RF 60 vs 120 | 59.65 | 18.51 | 50.68 | 54.75 | 83.75 | 22.83 |
| | RF 60 vs 150 | 99.30 | 47.62 | 88.03 | 93.14 | 129.59 | 53.04 |
| Interest rate (credit spread) **decreases** by 1.8 basis points (mean event between 2010-2018) | RF 60 vs 60 | -19.77 | -39.87 | -24.13 | -22.15 | -7.99 | -37.89 |
| | RF 60 vs 90 | 19.83 | -10.85 | 13.17 | 16.19 | 37.81 | -7.84 |
| | RF 60 vs 120 | 59.42 | 18.16 | 50.46 | 54.54 | 83.61 | 22.22 |
| | RF 60 vs 150 | 99.02 | 47.18 | 87.76 | 92.88 | 129.41 | 52.28 |
| Interest rate (credit spread) **decreases** by 24.6 basis points (all negative events) | RF 60 vs 60 | -21.28 | -42.17 | -25.52 | -23.53 | -8.95 | -41.86 |
| | RF 60 vs 90 | 17.52 | -14.36 | 11.05 | 14.08 | 36.34 | -13.89 |
| | RF 60 vs 120 | 56.31 | 13.45 | 47.62 | 51.69 | 81.63 | 14.08 |
| | RF 60 vs 150 | 95.11 | 41.26 | 84.19 | 89.31 | 126.92 | 42.05 |
| Interest rate (credit spread) **increases** by 26.4 basis points (all positive events) | RF 60 vs 60 | -18.19 | -37.36 | -22.62 | -20.65 | -7.00 | -33.75 |
| | RF 60 vs 90 | 22.24 | -7.03 | 15.47 | 18.48 | 39.32 | -1.52 |
| | RF 60 vs 120 | 62.67 | 23.30 | 53.57 | 57.61 | 85.63 | 30.72 |
| | RF 60 vs 150 | 103.10 | 53.64 | 91.66 | 96.74 | 131.95 | 62.96 |
| No. | | 6 (A)<br>59 (BBB) | 6 (A)<br>21 (BB) | 59 (BBB)<br>21(BB) | 59 (BBB)<br>13 (B) | 4 (Producer)<br>16 (Supplier) | 1 (Ford, BBB)<br>1 (Goodyear, BB) |

Notes: This Table reports the cost (positive sign) and benefits (negative sign) of reverse factoring from the point of view of a supplier corporation. The computation assumes a benchmark loan volume of 1.000.000 Euro. The profit margin of banks is assumed to be 50 basis points. The central bank interest rate is assumed to be zero. The computation is based on daily timeseries from 2010 to 2018 for 70 US-corporations in the category consumer non-durables. The last row represents the number of corporations in different rating categories. The reading of the Table is as follows: The row 'Benchmark' characterizes a benchmark constellation with different maturities. The supplier needs a loan of 60 days but the originator (producer) offers a reverse factoring contract. The payment term is ranging from 60 days (RF 60 vs 60) up to 150 days (RF 60 vs 150). The other rows reflect the cost-benefit trade-off of positive and negative events. All events are based on the event study. The column 'Rating A vs Rating BBB' denotes a situation where we have 59 suppliers with a BBB-rating and 6 originators with an A-rating. The number of -19.66 represents a benefit of -19.66 percent or, in other words, the cost saving of reverse factoring in comparison to a bank loan to the supplier corporation. Source: Authors computations.

## Appendix C. Break-Even Simulation

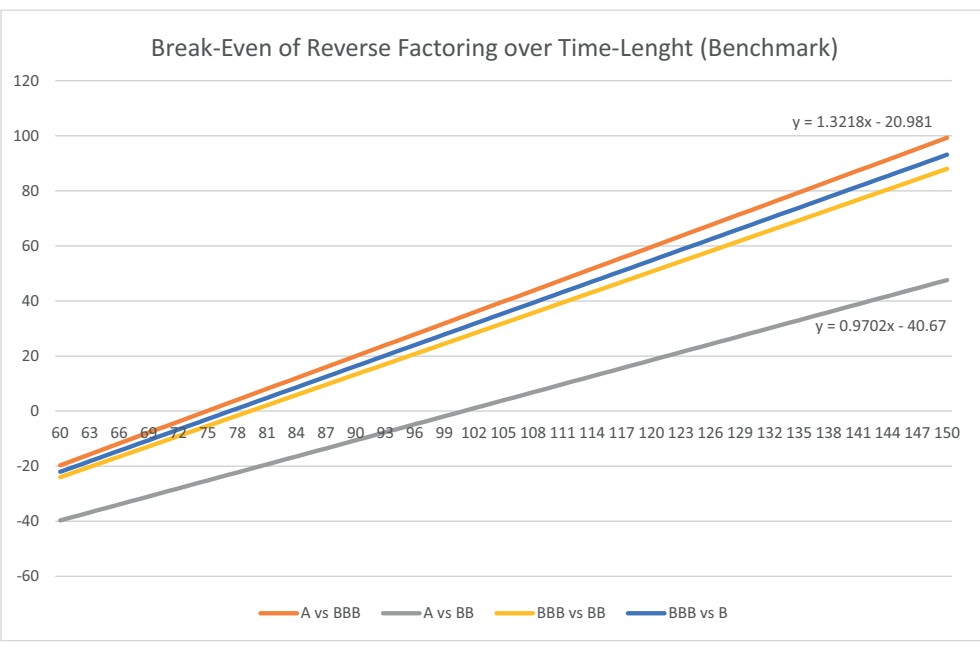

**Figure A1.** Break-even points. Source: Authors' simulation.

## Appendix D. Cost-Benefit Allocation over the Business Cycle

**Figure A2.** Reverse factoring over the business cycle. Source: Authors' simulation and estimation.

## Note

[1] Grüter and Wuttke (2017) indicate that, in many cases, buyers prefer payment term extensions in certain industries, even up to double payment terms, and this would still be beneficial for both parties.

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
