# Peer review of "Supply Chain Finance: Cost–Benefit Differentials under Reverse Factoring with Extended Payment Terms"

_ijfs, doi:10.3390/ijfs9040059_

Round 1
Reviewer 1 Report
Manuscript: Supply Chain Finance: Cost-Benefit Differentials under Reverse Factoring with extended Payment Terms
Structure and logic of manuscript; structure and logic research problems elaboration in the paper are correct.
Abstract; Very well structured with clear aims, scope of analysis, method applied and brief results.
Introduction and literature overview; rigorous relevant literature overview, with clearly derived research gaps and added value also with brief findings included.
Method of research; clear assumptions and rigorous elaboration of the two-stage simulation model .
General Results and Discussions; in-depth and multi-financial aspects elaborated; findings clearly discussed with limitations of study included.
Conclusions; if the presented content structure between General Results and Discussions and Conclusions are maintained, then general conclusions are relevant and sufficient.
Summing-up I have no major or minor remarks to this highly valuable manuscript and appreciate proficient writing style and paper editing and recommend this paper for publishing.
I congratulate the Authors for a very good job and rigorous research of the important and complex problem. It is highly recommended to continue the research.c
Author Response
Referee 1:
Thank you very much for your encouraging feedback – this is highly appreciated!
We will make little adjustments according to minor feedback by the second referee.
Yet, we have enjoyed reading your feedback and your judgement that the paper is already ready for publication.
Please find the revised manuscript attached with all revisions marked up in red.
Reviewer 2 Report
Dear Authors,
Thank you for your manuscript, it is exciting, but I have some comments:
In my opinion, the introduction part of your paper is too long; some information you can transfer to the literature review.
The literature review looks good, but it will be nice if you add and the newest scientific works from the period: 2020-2021. Only two references are 2019. Please renew this part.
The methodology part - is good.
Please separate the result and discussion parts.
And please rewrite the conclusion.
Author Response
Dear Referee 2:
Thank you very much for your valuable and constructive feedback – this is highly appreciated!
We have taken up your feedback and confirm that our paper has improved by your comments. We made the following adjustments:
ad 1. Your major comment was about including some recent literature from 2020 or 2021. Indeed, we have included recently published articles in relation to our topic and field of research in the Literature review. The newly added research work was all published in 2020 and 2021:
- Bal, M., Pawlicka, K., Supply Chain Finance and Challenges of Modern Supply Chains. Scientific Journal of Logistics 2021, 17(1), 71–82.
- Gupta, N., Soni, G., A Decision-Making Framework for Sustainable Supply Chain Finance in Post-COVID Era. International Journal of Global Business and Competitivness 2021, 1–10.
- Lam, H.K.S, Zhan, Y., The impacts of supply chain finance initiatives on firm risk: evidence from service providers listed in the US. International Journal of Operations & Production Management 2021, 41(4), 383–409.
- Lampón, J.F., Relevance of the cooperation in financing the automobile industry’s supply chain: the case of reverse factoring. Journal of Manufactoring Technology Management 2021, 32(5), 1094–1112.
- Liu, Z., Literature Review of Supply Chain Finance Based on Blockchain Perspective. Open Journal of Business Management 2021, 9, 419–429.
- Marchi, B., Zanomi, S., Jaber, M.Y., Improving Supply Chain Profit through Reverse Factoring: A New Multi-Suppliers Single-Vendor Joint Economic Lot Size Model. International Journal of Financial Studies 2020, 8(23), 1-16.
- Wiedmer, R., Griffis, S.E., Structural characteristics of complex supply chain networks. Journal of Business Logistics 2021, 42, 264–290.
- Wiedmer, R., Rogers, Z.S., Polyviou, M., Mena, C., Chae, S., The Dark and Bright Sides of Complexity: A Dual Perspective on Supply Network Resilience Journal of Business Logistics 2021, 42(3), 336–359.
Of course, we added the additional references as you have recommended.
Ad 2. We have considered your recommendation to separating results and discussion. Yet, after long discussion, we felt that given the nature of our study approach, a separation of result and discussion might potentially imply redundancies and might not add value to the paper. Therefore, we separate results and discussions more clearly by utilizing different paragraphs and indicate by keywords (see red color) at the beginning of each paragraph, which part is about the findings and results and which paragraph is a discussion with managerial conclusions. We are strongly convinced that this structure fits best to our approach and paper.
Ad. 3. Finally, as recommended, we shortened the Introduction section by moving a paragraph to the literature review. In addition, we rewrote the conclusion section as well as made a few refinements on language and spelling checks.
Please find the revised manuscript attached with the revisions marked up in red color.
Thank you for letting us know if you have any further comments or questions.

Round 2
Reviewer 2 Report
Dear Authors,
thank you for your improvements. Good luck.
Author Response
Thank you for your excellent feedback.